# Efficacy, Safety, and Immunogenicity of Subunit Respiratory Syncytial Virus Vaccines: Systematic Review and Meta-Analysis of Randomized Controlled Trials

**DOI:** 10.3390/vaccines12080879

**Published:** 2024-08-02

**Authors:** Yuhang Wu, Yuqiong Lu, Yuwei Bai, Bingde Zhu, Feng Chang, Yun Lu

**Affiliations:** School of International Pharmaceutical Business, China Pharmaceutical University, Nanjing 211198, China; wuyh529@outlook.com (Y.W.); luyuqiong96@foxmail.com (Y.L.); qw1033766671@163.com (Y.B.); 15165023113@163.com (B.Z.)

**Keywords:** efficacy, immunogenicity, safety, respiratory syncytial virus vaccines, meta-analysis

## Abstract

Background: Respiratory syncytial virus (RSV) is garnering increasing attention, with a growing number of subunit RSV vaccines under active clinical investigation. However, comprehensive evidence is limited. Methods: We conducted a comprehensive search across PubMed, Embase, the Cochrane Library, Web of Science, and ClinicalTrials.gov from database inception to 12 January 2024, focusing on published randomized controlled trials (RCTs). Results: A total of 17 studies were included, encompassing assessments of efficacy (5 studies), safety (17 studies), and immunogenicity (12 studies) of RSV subunit vaccines. The pooled risk ratio (RR) for RSV-associated acute respiratory infection (RSV-ARI) with subunit vaccines was 0.31 (95% CI: 0.23–0.43), for RSV-associated lower respiratory tract infection (RSV-LRTI), it was 0.32 (95% CI: 0.22–0.44), and for severe RSV-LRTI (RSV-SLRTI), it was 0.13 (95% CI: 0.06–0.29). There was no significant difference in serious adverse events (SAEs) between the vaccine and placebo groups, with a pooled RR of 1.05 (95% CI: 0.98–1.14). The pooled standardized mean difference (SMD) for the geometric mean titer (GMT) of neutralizing antibodies was 2.89 (95% CI: 2.43−3.35). Conclusion: Subunit RSV vaccines exhibit strong efficacy, favorable safety profiles, and robust immunogenicity. Future research should focus on the cost-effectiveness of various vaccines to enhance regional and national immunization strategies.

## 1. Introduction

Respiratory syncytial virus (RSV) is a widespread respiratory pathogen that affects individuals of all ages, with significant consequences for vulnerable groups such as young children, the elderly, and those with weakened immune systems [1,2]. RSV is a leading cause of acute lower respiratory tract infections. Recent data from 2019 indicate that around 33 million children under five worldwide were affected by acute lower respiratory tract infections due to RSV, resulting in 3.6 million hospitalizations and 26,300 in-hospital deaths [3]. Furthermore, research by Savic (2023) highlights that in 2019, RSV was responsible for 470,000 hospitalizations and 33,000 deaths among the elderly population aged 60 and above in high-income countries [4]. The substantial burden of disease imposed by RSV is evident. Given that the majority of RSV-related fatalities occur outside hospital settings, the actual burden is likely underreported [5,6].

Currently, there are no specific antiviral therapies available for the treatment of RSV infection, with post-infection management relying primarily on supportive care [7]. Therefore, the prevention of RSV is significant. Preventive measures against RSV include the administration of monoclonal antibody medications and vaccines. Palivizumab and nirsevimab are the monoclonal antibodies currently approved for RSV prevention, though their use is restricted to infants. However, the high cost of palivizumab presents a substantial financial burden for low- and middle-income families, and the cost-effectiveness of nirsevimab remains to be established [8]. Consequently, in light of the limitations associated with monoclonal antibodies, the development of vaccines targeting RSV has become increasingly crucial.

RSV candidate vaccines utilize various strategies, including recombinant vector vaccines, subunit vaccines, particle vaccines, live attenuated vaccines, chimeric vaccines, and nucleic acid vaccines [9]. Among these, subunit vaccines were the first to reach clinical application. In May and June 2023, the FDA approved two subunit vaccines for preventing RSV infection in older adults: Arexvy (GlaxoSmithKline, London, UK) and Abrysvo (Pfizer Inc., New York, USA). Furthermore, Abrysvo has also been approved for preventing RSV in infants and young children. Both vaccines have been endorsed by the Advisory Committee on Immunization Practices (ACIP) for use in individuals aged 60 and older [10].

Simoes EA (2001) [11] conducted the first systematic review on subunit RSV vaccines, focusing on the safety data of the purified F protein subunit (PFP) vaccine [11]. Shan J (2021) [12] provided a descriptive summary of the safety and immunogenicity data of RSV candidate vaccines, finding that subunit vaccines demonstrated good safety profiles and elicited strong immune responses [12]. Currently, no studies have analyzed the efficacy of subunit vaccines, and there have been no quantitative evaluations of various indicators. With the increasing number of subunit vaccines entering RCTs, further evidence of their efficacy, safety, and immunogenicity in preventing RSV infection is necessary.

This study aims to conduct a systematic review and meta-analysis of the efficacy, safety, and immunogenicity of subunit vaccines in preventing RSV infection. RCTs will be analyzed to compare vaccinated individuals with placebo controls. Efficacy will be measured using the RR for RSV-related infections, immunogenicity using the SMD for neutralizing antibody titers, and safety by the incidence of adverse events. This comprehensive evaluation is essential for healthcare professionals to understand the overall benefits and risks associated with these vaccines, enabling them to make evidence-based decisions and optimize immunization strategies for better public health outcomes.

## 2. Methods

The protocol for this study has been registered with PROSPERO under the registration number CRD42024524720. This study adheres to the Preferred Reporting Items for Systematic Reviews and Meta-Analyses (PRISMA) checklist for systematic reviews and meta-analyses [13].

### 2.1. Search Strategy

For this systematic review and meta-analysis, a comprehensive search was conducted across PubMed, Embase, the Cochrane Library, Web of Science, and ClinicalTrials.gov, covering all studies published up to 12 January 2024. We utilized a combination of six terms and their synonyms, using both Medical Subject Headings (MeSHs) and free-text terms: respiratory syncytial virus, respiratory tract infection, vaccine, vaccination, subunit vaccine, and randomized controlled trial. The search strategy for PubMed, which can be adapted for other databases, is outlined in Appendix A.

### 2.2. Study Selection

The inclusion and exclusion criteria are based on the PICOS principles. The inclusion criteria are as follows: (1) no restrictions on the study population; (2) the intervention is a subunit respiratory syncytial virus vaccine; (3) the control measure is placebo; (4) studies that include relevant outcome measures of RSV vaccine efficacy, safety, or immunogenicity; (5) RCTs.

The exclusion criteria are defined as follows: (1) studies that involve the combined use of other vaccines; (2) non-original research such as conference abstracts, reviews, meta-analyses, etc.; (3) studies conducted on animal or in vitro models and cell lines; (4) studies from which data could not be extracted despite efforts to contact the authors; (5) duplicate publications; (6) studies published in languages other than English.

### 2.3. Data Extraction and Outcome Measures

The study involved two researchers (BYW and ZBD) who independently extracted the data. Any disagreements were discussed and resolved with a third researcher (WYH). We used Microsoft Excel to construct a standardized data extraction form to record the extracted information, including (1) basic information: first author and publication year; (2) study characteristics: country/region, sample size, blinding method, study phase, and funding sources; (3) participant information: age range and gender ratio; (4) intervention information: the name of the vaccine in the intervention group, dosage, and whether the vaccine contained adjuvants; (5) outcome information: relevant outcomes of efficacy, safety, and immunogenicity, as well as follow-up time.

For dose exploration studies with multiple intervention groups, we included only those using the standard dose, defined as the dose of an approved vaccine product or the dose selected for phase III trials. According to WHO Preferred Product Characteristics for Respiratory Syncytial Virus Vaccines and the frequency of current mainstream vaccines, we only included outcome events after the first dose of the vaccine. As neutralizing antibody levels typically peak approximately one month after receiving a subunit vaccine, we considered only the neutralizing antibody data one month post-vaccination. When detailed information on outcomes was incomplete, we contacted the study authors directly or used data extraction software (GetData Graph Digitizer 2.22) to extract data from the original literature.

Primary outcome measures included the following: (1) efficacy, defined as RSV acute respiratory infection (RSV-ARI), acute lower respiratory tract infection (RSV-LRTI), and severe acute lower respiratory tract infection (RSV-SLRTI); (2) immunogenicity, assessed by neutralizing antibody titers; and (3) vaccine safety, evaluated by serious adverse events. Secondary outcome measures included the analysis of any local adverse events, any systemic adverse events, and specific adverse events such as redness at the injection site, pain at the injection site, swelling at the injection site, fever, headaches, fatigue, muscle pain, joint pain, nausea or vomiting, and chills.

### 2.4. Quality Assessment

We used the Cochrane Collaboration tool (RoB 2) to assess the risk of bias for randomized trials [14]. This tool evaluates several types of biases, including selection bias, performance bias, detection bias, attrition bias, and reporting bias, which helps in identifying potential sources of bias that might affect the validity of the trial results. The quality of evidence for the primary outcomes was evaluated using the Grade of Recommendation, Assessment, Development, and Evaluation (GRADE) system [15]. The GRADE system assesses evidence quality across five domains: study limitations (risk of bias), imprecision (the degree of certainty around the effect estimate), inconsistency (variability in results across studies), indirectness (differences in study population, intervention, comparator, or outcomes), and publication bias (selective publication of studies). These tools were chosen because they provide a systematic and transparent framework for evaluating the reliability and relevance of evidence from clinical trials, ensuring that the conclusions drawn are well supported and credible.

### 2.5. Data Analysis

This meta-analysis was performed using Stata 18.0 software. Efficacy and safety outcomes, being binary variables, were expressed as relative risk (RR) with 95% confidence intervals (CIs). Studies with no events in both the vaccination and control groups were excluded from the RR summary [16]. For studies where only one group had no events, a fixed value of 0.5 was added to each cell of the 2 × 2 table to correct for continuity [16]. Heterogeneity among the included studies was assessed using the I^2^ statistic. A random-effects model was applied when I^2^ exceeded 50%, whereas a fixed-effects model was utilized when I^2^ was 50% or less.

Immunogenicity outcomes are continuous variables with different units; hence, we used the standardized mean difference (SMD) of log-transformed geometric mean titers (GMTs) and the corresponding 95% CIs to represent the difference in immunogenicity between the vaccination and control groups. The SMD was calculated based on the DerSimonian–Laird model using the means and standard deviations of the two groups, with missing standard deviations estimated from the confidence intervals and sample sizes [16].

This study planned to conduct subgroup analyses based on age demographics (older adults, adults, children) and the inclusion of adjuvants in vaccines to identify potential heterogeneity sources. We utilized funnel plot asymmetry and the Egger test to evaluate potential publication bias. However, the publication bias test was omitted when fewer than 10 studies were included. Additionally, sensitivity analyses were carried out to assess the influence of individual studies on the overall estimates. In this study, a *p*-value of less than 0.05 was regarded as statistically significant.

## 3. Results

### 3.1. Search Results and Study Characteristics

We identified a total of 14,872 records, of which 3231 were duplicates and subsequently excluded. After screening the titles and abstracts of the remaining 11,641 studies, 71 studies met the inclusion criteria. The studies excluded after the full-text screening and the reasons for their exclusion are detailed in Appendix A. Following a full-text review, data from 17 studies were included in the analysis (Figure 1) [17,18,19,20,21,22,23,24,25,26,27,28,29,30,31,32,33].

Table 1 summarizes the basic characteristics of the included studies. Overall, eight studies were multicenter trials that included participants with diverse age characteristics; six studies of subunit vaccines contained adjuvants, while eleven did not. All eligible studies were phase I/II to phase III RCTs; no phase IV studies were reported. Efficacy data were reported in five studies, vaccine safety was addressed in seventeen studies, and twelve studies provided information on vaccine immunogenicity.

### 3.2. Quality Assessment

For the three different efficacy outcomes, eight studies were assessed as having a moderate risk of bias, and one study was assessed as having a low risk of bias. For the immunogenicity outcomes, one study was rated as high risk, seven studies as moderate risk, and four studies as low risk. For the safety outcomes, one study was rated as high risk, three studies as moderate risk, and eight studies as low risk (Appendix A).

### 3.3. The Efficacy of RSV Subunit Vaccines

Three studies were included to evaluate the efficacy of RSV-ARI [20,27,33], as shown in Figure 2. The overall impact of vaccination on acute respiratory infections favored the vaccine group, with a combined risk ratio of 0.31 (95% CI: 0.23–0.43, I^2^ = 0%), and no statistically significant heterogeneity was observed (*p* = 0.612). According to the GRADE system, the quality of evidence for the vaccine’s efficacy against RSV-ARI is considered moderate (Appendix A).

Four studies were included to evaluate the efficacy of RSV-LRTI [21,27,31,33], as shown in Figure 3. The results show that the overall impact of vaccination on lower respiratory tract infections favored the vaccine group, with a combined risk ratio of 0.32 (95% CI: 0.22–0.44, I^2^ = 0%), and no statistically significant heterogeneity was observed (*p* = 0.200). According to the GRADE system, the quality of evidence for the vaccine’s efficacy against RSV-LRTI is considered moderate (Appendix A).

Three studies were included to evaluate the efficacy of RSV-SLRTI [21,27,31], as shown in Figure 4. The results show that the overall impact of vaccination on severe lower respiratory tract infections favored the vaccine group, with a combined risk ratio of 0.13 (95% CI: 0.06–0.29, I^2^ = 0%), and no statistically significant heterogeneity was observed (*p* = 0.535). According to the GRADE system, the quality of evidence for the vaccine’s efficacy against RSV-SLRTI is considered moderate (Appendix A).

Subgroup and sensitivity analyses were performed for the three efficacy outcomes. The subgroup analyses were based on the age of the subjects and the presence of adjuvants in the vaccine. While the results differed from the original outcomes, no statistically significant differences were noted (*p* > 0.05), as shown in Appendix A. The sensitivity analysis revealed that no individual study significantly impacted the results, indicating the robustness of the findings, as shown in Appendix A.

### 3.4. The Safety of RSV Subunit Vaccines

All 17 studies assessed safety outcomes, including any local adverse events, systemic adverse events, and serious adverse events. The risk of experiencing any local adverse events (RR = 3.58, 95% CI: 2.00–6.41) and any systemic adverse events (RR = 1.40, 95% CI: 1.05–1.85) was significantly higher in the vaccine group compared to the placebo group (Table 2). Specifically, the risk of local adverse events (such as pain, redness, and swelling at the injection site) in the vaccine group (RR = 3.75–5.49) exceeded that of systemic adverse events (RR = 1.05–1.91). Only one study reported the occurrence of chills post-vaccination, with an incidence of 6/100 in the vaccine group and 3/101 in the placebo group, thus precluding a meta-analysis [26].

The risk of serious adverse events is a key focus for vaccine safety. Twelve studies were included to assess serious adverse events following vaccination [17,19,20,21,22,26,27,29,30,31,32,33], as shown in Figure 5. The results show no significant difference in the incidence of serious adverse events between the vaccine group and the placebo group (RR = 1.05, 95% CI: 0.98–1.14). According to the GRADE system, the quality of evidence for serious adverse events is considered moderate (Appendix A).

Subgroup analyses were performed considering the age characteristics of the participants and the presence of adjuvants in the vaccine. The results align with the overall findings, showing no statistically significant differences across subgroups (*p* > 0.05), as depicted in Appendix A. The sensitivity analysis confirmed the robustness of the results, with no significant publication bias detected by the funnel plot and Egger’s test (*p* = 0.929), as illustrated in Appendix A.

### 3.5. Immunogenicity of RSV Subunit Vaccines

Twelve studies were included to evaluate immunogenicity following vaccination [17,20,22,23,25,26,27,28,30,31,32], as shown in Figure 6. The standardized mean difference (SMD) for the geometric mean titers (GMTs) of neutralizing antibodies post-vaccination was 2.89 (95% CI: 2.43−3.35). According to the GRADE system, the quality of evidence for neutralizing antibodies is considered low (Appendix A).

The subgroup analysis showed that elderly subjects produced higher levels of neutralizing antibodies compared to the other two groups (SMD = 3.6), and the subgroup of vaccines containing adjuvants generated higher levels of neutralizing antibodies compared to those without adjuvants (SMD = 4.24), as shown in Appendix A. The sensitivity analysis indicated that no single included study significantly affected the immunogenicity results. The slightly asymmetrical funnel plot and Egger’s test did not reveal significant publication bias (*p* = 0.134), as illustrated in Appendix A.

## 4. Discussion

This systematic review and meta-analysis provides evidence on the efficacy, safety, and immunogenicity of subunit vaccines targeting respiratory syncytial virus. Our findings indicate that the following: (1) subunit vaccines are highly effective in preventing severe RSV-associated lower respiratory tract infections, with a pooled relative risk of 0.13 (Figure 4); (2) there is no significant difference in the incidence of serious adverse events between vaccine recipients and placebo groups, with a pooled RR of 1.05 (Figure 5); (3) subunit vaccines markedly enhance neutralizing antibody levels in participants, with a standardized mean difference of 2.89 (Figure 6). The subgroup analyses based on age and adjuvant inclusion showed consistent results, and the sensitivity analyses confirmed that excluding any single study did not change the overall findings. These analyses support the robustness of the results, suggesting that subunit vaccines are viable candidates for clinical application, albeit with careful consideration of local adverse reactions induced by the vaccines (RR = 3.75–5.49) (Table 2).

Currently, there is no consensus on the standardized methods for evaluating RSV vaccine efficacy endpoints, leading to variability in outcome measures across different RCTs [34]. Therefore, we included three different outcome indicators—acute respiratory infections (ARIs), lower respiratory tract infections (LRTIs), and severe lower respiratory tract infections (SLRTIs)—to comprehensively assess the efficacy of the vaccine. In a previous study that used patient-reported outcomes to confirm vaccine endpoint efficacy, ARIs and LRTIs were considered reliable outcome indicators [35]. Additionally, these three indicators have been recognized by the WHO. Our analysis revealed that subunit vaccines achieved an 87% efficacy in preventing severe RSV infections, surpassing the WHO’s benchmark of 70% effectiveness [36]. The efficacy against RSV-induced ARIs and LRTIs was 69% and 68%, respectively, closely aligning with the expected effectiveness. These findings underscore the potential of subunit vaccines in effectively preventing RSV infections.

Regarding vaccine safety, subunit vaccines are linked to a higher risk of local and systemic adverse events; however, there is no significant difference in the incidence of serious adverse events compared to the placebo group, consistent with previous studies [11,12]. The subgroup analysis also indicated no significant differences in the incidence of serious adverse events across different age groups, suggesting that the safety profile of subunit vaccines is acceptable.

Regarding immunogenicity, the failure of RSV vaccine development has often been attributed to insufficient immunogenicity [33]. In this study, we selected neutralizing antibody titers as the primary immunogenicity indicator, as research has demonstrated that the level of neutralizing antibodies is a critical determinant of vaccine efficacy [37]. Our findings indicate that subunit vaccines significantly increase neutralizing antibody levels compared to placebo. However, there was considerable heterogeneity in the results, potentially due to several factors. Firstly, the varying quality of studies may significantly influence the outcomes (one study on neutralizing antibody outcomes was rated as high, and seven were rated as moderate). Secondly, previous immune responses should be taken into account when assessing the immunogenicity of adult RSV vaccines, as adults are likely to have been previously exposed to RSV, resulting in baseline titer differences. Unfortunately, few RCTs stratify participants based on prior RSV infection, precluding analysis of this factor. We recommend that future RCTs account for baseline differences among subjects to provide a more accurate evaluation.

Vaccination is a highly effective measure to reduce disease burden. Beyond efficacy and safety, the economic aspect is also a critical factor to consider. Given that most RSV-related deaths occur in low- and middle-income countries, promoting RSV vaccines in these regions is particularly important. However, several barriers exist, including insufficient awareness of RSV, a lack of relevant national surveillance data, and the high costs associated with RSV diagnostic monitoring and related vaccine products [38]. Consequently, future efforts should focus on conducting cost-effectiveness evaluations for different vaccines to enhance immunization programs across various countries and regions.

This study encompasses not only the two subunit vaccines already available on the market but also those currently undergoing RCTs. In terms of target populations, besides the primary groups for RSV vaccine development—children, pregnant women, and the elderly—it is also essential to include data from healthy adults who have received the vaccine. This comprehensive approach aims to provide evidence for potentially expanding the vaccination population in the future, thereby further promoting herd immunity, which is crucial for the prevention of infectious diseases.

However, several potential limitations should be noted. Firstly, the number of studies included for combining different vaccine efficacy outcomes was relatively small, so the combined results should be interpreted with caution. Secondly, the evaluation methods for vaccine immunogenicity outcomes, such as neutralizing antibody titers and T cell responses, were varied. We did not evaluate T cell responses post-vaccination, primarily because few studies reported data on this aspect. Thirdly, different studies employed different methods to detect neutralizing antibody levels, potentially leading to high heterogeneity. Lastly, some of the included studies (phase I/II trials) had a small number of participants, which may result in the safety outcomes of the vaccine being either underestimated or overestimated.

## 5. Conclusions

This meta-analysis shows that subunit vaccines are effective, safe, and boost immune responses. They significantly reduce severe RSV infections without increasing serious adverse events. These findings support the inclusion of subunit RSV vaccines in immunization programs, with attention paid to local adverse reactions. Future studies should explore cost-effectiveness for wider use.

## Figures and Tables

**Figure 1 vaccines-12-00879-f001:**
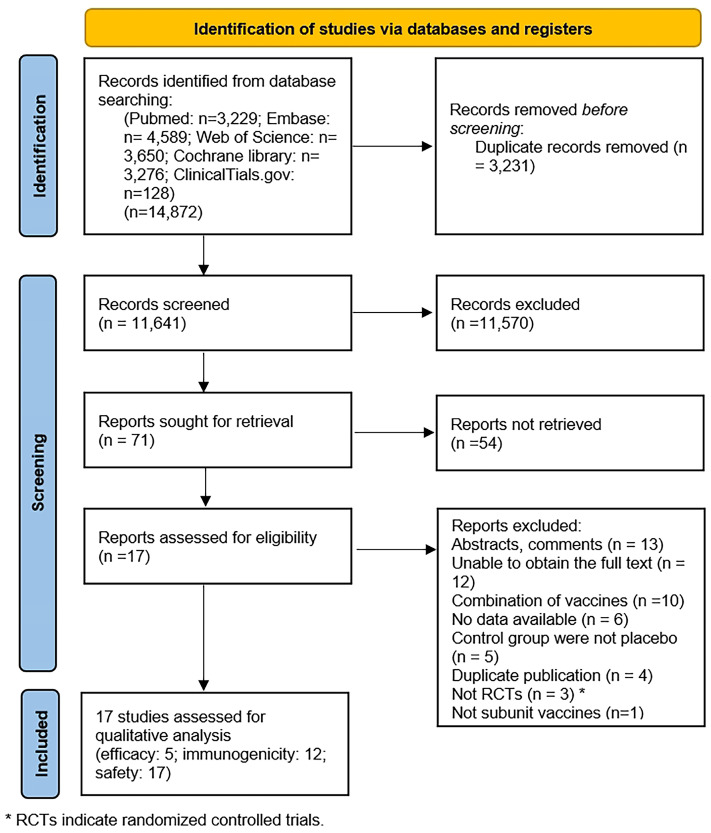
PRISMA flow diagram.

**Figure 2 vaccines-12-00879-f002:**
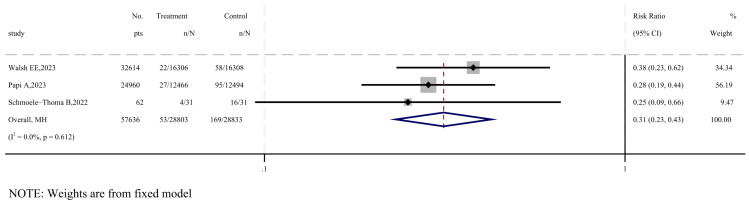
Forest plot of efficacy in vaccine and placebo groups of RSV-ARI [27,28,33].

**Figure 3 vaccines-12-00879-f003:**
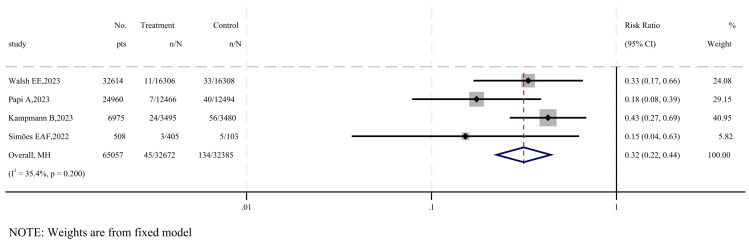
Forest plot of efficacy in vaccine and placebo groups of RSV-LRTI [21,27,31,33].

**Figure 4 vaccines-12-00879-f004:**
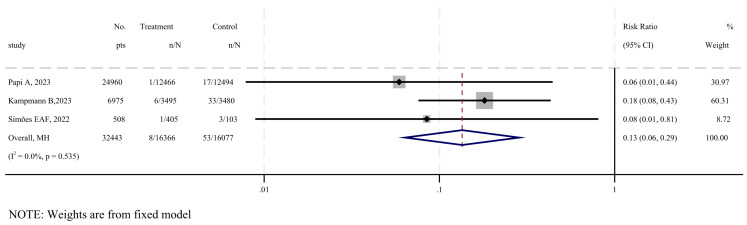
Forest plot of efficacy in vaccine and placebo groups of RSV-SLRTI [21,27,31].

**Figure 5 vaccines-12-00879-f005:**
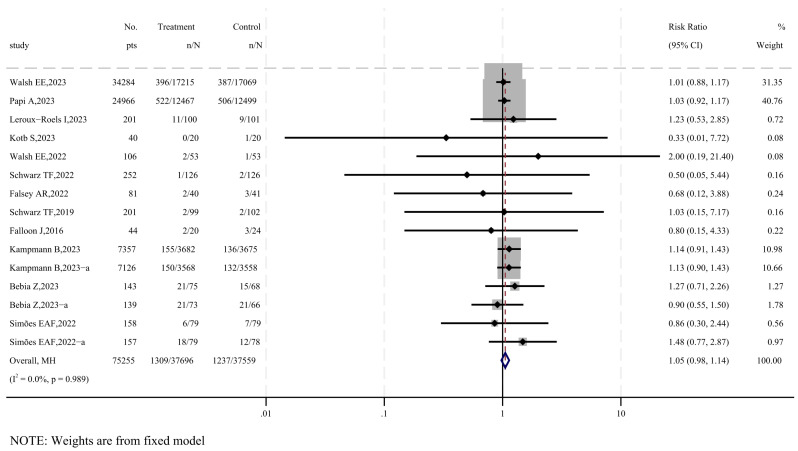
Forest plot of SAEs in vaccine and placebo groups [17,19,20,21,22,26,27,29,30,31,32,33].

**Figure 6 vaccines-12-00879-f006:**
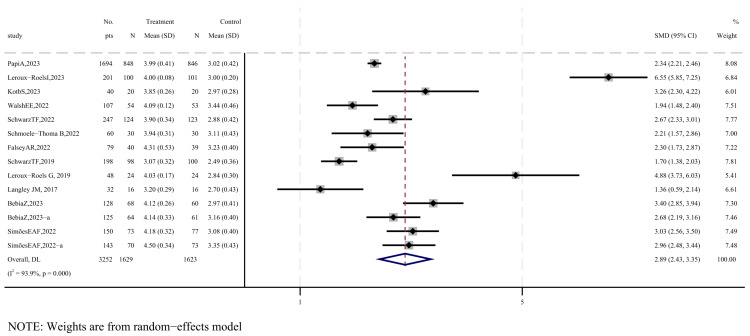
Forest plot of GMT values of log-transformed neutralizing antibodies [17,20,22,23,25,26,27,28,29,30,31,32].

**Table 1 vaccines-12-00879-t001:** Summary of research articles.

No	First Author, Year	Country	Phase	Target Population; Age	Sample Size	Sex (% Female)	Vaccine Name	Adjuvant	Efficacy			Safety		Immunogenicity
									RSV-ARI	RSV-LRTI	RSV-SRLI	Local or System AEs	Serious AEs	GMT
1	Bebia Z, 2023 [17]	multicenter	phase 2	Pregnant Women; 18–40 y	143	100	RSVPreF3	None	NR	NR	NR	YES	YES	YES
2	Cheng X, 2023 [18]	Australia	phase 1	Adults; 18–45 y	60	76.7	BARS13	cyclosporine A	NR	NR	NR	YES	YES	NR
3	Falloon J, 2016 [19]	United States	phase 1a	Old Adults; ≥60 y	44	51.2	RSV sF	glucopyranosyl lipid A	NR	NR	NR	YES	YES	NR
4	Falsey AR, 2022 [20]	United States	phase 1/2	Old Adults; 65–85 y	81	63	RSVpreF	None	NR	NR	NR	YES	YES	YES
5	Kampmann B, 2023 [21]	multicenter	phase 3	Pregnant Women; 18–49 y	7392	100	RSVpreF	None	NR	YES	YES	YES	YES	NR
6	Kotb S, 2023 [22]	Japan	phase 1	Old Adults; 60–80 y	40	50	RSVPreF3	AS01 E	NR	NR	NR	YES	YES	YES
7	Langley JM, 2017 [23]	Canada	phase 1	Adults; 18–44 y	32	0	RSV-PreF	None	NR	NR	NR	YES	YES	YES
8	Langley JM, 2018 [24]	Canada	phase 1	Old Adults; 50–64 y	16	72.5	DPX-RSV	None	NR	NR	NR	YES	YES	NR
9	Leroux-Roels G, 2019 [25]	Belgium	phase 1	Adults; 18–45 y	48	66.7	RSV F	Al(OH)3	NR	NR	NR	YES	YES	YES
10	Leroux-Roels I, 2023 [26]	multicenter	phase 1/2	Old Adults; 60–80 y	201	57.2	RSVPreF3	AS01 E	NR	NR	NR	YES	YES	YES
11	Papi A, 2023 [27]	multicenter	phase 3	Old Adults; ≥60 y	24,966	51.7	RSVpreF3	AS01 E	YES	YES	YES	YES	YES	YES
12	Schmoele-Thoma B, 2022 [28]	United Kingdom	phase 2a	Adults; 18–50 y	70	29	RSVpreF	None	YES	NR	NR	YES	YES	YES
13	Schwarz TF, 2022 [29]	multicenter	phase 1/2	Adults; 18–45 y	252	100	RSVPreF3	None	NR	NR	NR	YES	YES	YES
14	Schwarz TF, 2019 [30]	multicenter	phase 2	Adults; 18–45 y	201	100	RSVpreF	None	NR	NR	NR	YES	YES	YES
15	Simões EAF, 2022 [31]	multicenter	phase 2b	Pregnant Women; 18–49 y	158	100	RSVpreF	None	NR	YES	YES	YES	YES	YES
16	Walsh EE, 2022 [32]	United States	phase 1/2	Adults; 18–49 y	107	63.6	RSVpreF	None	NR	NR	NR	YES	YES	YES
17	Walsh EE, 2023 [33]	multicenter	phase 3	Old Adults; ≥60 y	34,284	49.2	RSVpreF	None	YES	YES	NR	YES	YES	NR

RSV: respiratory syncytial virus; ARI: acute respiratory infection; LRTI: lower respiratory tract infection; SLRTI: severe lower respiratory tract infection; GMT: geometric mean titer; AE: adverse event; YES: this study reported on the relevant outcomes; NR: this study did not report on the relevant outcomes; y: year.

**Table 2 vaccines-12-00879-t002:** The incidence of adverse events among the intervention group versus the control group.

Adverse Events	No. of Studies	Reactions/Total		RR (95% CI)	Heterogeneity I^2^ (%)	p-Heterogeneity
		Intervention	Control			
Local adverse events (any)	7	1932/5625	974/5543	3.58 (2.00, 6.41)	97.5	0.000
Systemic adverse events(any)	7	1665/5625	1400/5534	1.40 (1.05, 1.85)	91.0	0.000
Pain	16	2320/8895	664/8772	4.00 (2.92, 5.49)	82.9	0.000
Redness	13	465/8851	83/8724	5.49 (4.36, 6.91)	42.1	0.055
Swelling	12	334/7957	76/7837	3.75 (1.90, 7.38)	60.9	0.003
Fatigue	15	2780/8887	2317/8764	1.23 (1.05, 1.44)	67.3	0.000
Headaches	15	2097/8887	1685/8764	1.37 (1.08, 1.74)	86.6	0.000
Myalgia	11	1447/7706	943/7585	1.52 (1.41, 1.63)	33.6	0.130
Nausea	9	923/8510	882/8774	1.05 (0.97, 1.15)	48.9	0.048
Pyrexia	10	333/8598	207/8479	1.91 (1.10, 3.32)	75.5	0.000
Arthralgia	7	488/4022	428/3981	1.14 (1.01, 1.29)	12.6	0.334

RR: risk ratio; CI: confidence interval.

## Data Availability

The data presented in this study are available in the article.

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
