# Peer review of "Efficacy, Safety, and Immunogenicity of Subunit Respiratory Syncytial Virus Vaccines: Systematic Review and Meta-Analysis of Randomized Controlled Trials"

_vaccines, 2024, doi:10.3390/vaccines12080879_

Round 1

Reviewer 1 Report

Comments and Suggestions for Authors

Reviewer 2 Report

Comments and Suggestions for Authors

Manuscript titled "Efficacy, Safety and Immunogenicity of Subunit Respiratory Syncytial Virus Vaccines: A Systematic Review and Meta-analysis of Randomized Controlled Trials", by Wu et al, is self-explanatory on the intent and content. The manuscript is well written and to the point relevant to the results obtained with adequate background to highlight the need for this study.

The methodology (PRISMA) followed eliminates and narrows down the sample size during the screening process, pls provide explanation/details on the records excluded.

The table 1 needs to be submitted in a landscape orientation, current form is inconvenient to read. Table legend needs more explanation.

The text in the figures is not very clear to read. The figures are a mix of table and graph, can this be made more legible.

There should be a section in discussion on the limitations of methodology and finding of this study.

Comments on the Quality of English Language

NA
